# Optimizing the Integration of Microwave Processing and Enzymatic Extraction to Produce Polyphenol-Rich Extracts from Olive Pomace

**DOI:** 10.3390/foods12203754

**Published:** 2023-10-12

**Authors:** Gabriela A. Macedo, Paula de P. M. Barbosa, Fernanda F. G. Dias, Lauren M. Crawford, Selina C. Wang, Juliana M. L. N. De Moura Bell

**Affiliations:** 1Bioprocesses Laboratory, DEPAN/FEA (School of Food Engineering), Unicamp (University of Campinas), R. Monteiro Lobato, 80, Campinas 13083970, Brazilpaulapmb@unicamp.br (P.d.P.M.B.); 2Department of Food Science and Technology, University of California, Davis, One Shields Avenue, Davis, CA 95616, USA; 3Blount Fine Foods, Fall River, MA 02720, USA; 4Biological and Agricultural Engineering, University of California, Davis, One Shields Avenue, Davis, CA 95616, USA

**Keywords:** green extraction, enzymes, olive pomace, phenolic compounds

## Abstract

The integration of green technologies such as microwave- and enzyme-assisted extraction (MEAE) has been shown to improve the extraction efficiency of bioactive compounds while reducing processing time and costs. MEAE using tannase alone (MEAE-Tan), or in combination with cellulase and pectinase (MEAE-Tan-Cel-Pec), was optimized to produce enriched phenolic and antioxidant extracts from olive pomace. The individual and integrated impact of enzyme concentration, temperature, and pomace/water ratio were determined using a central composite rotatable design. Optimal extraction conditions for MEAE-Tan (60 °C, 15 min, 2.34% of enzyme (*w*/*w*), and 1:15 pomace/water ratio) and MEAE-Tan-Cel-Pec (46 °C, 15 min, 2% of enzymes (*w*/*w*), in the proportion of 1:1:1, and 1:20 pomace/water ratio) resulted in extracts containing 7110.6 and 2938.25 mg GAE/kg, respectively. The antioxidant activity of the extracts was correlated with phenolic acid release, which was enzyme-dependent, as determined with HPLC-DAD analysis. Enzyme selection had a significant impact on the phenolic profile of extracts, with tannase releasing high concentrations of chlorogenic acid and the combined use of enzymes releasing high concentrations of hydroxytyrosol and chlorogenic and ferulic acids. The novelty of this study relies on the integration and optimization of two green technologies (microwave- and enzyme-assisted extraction) to improve the extraction efficiency of bioactive phenolics from olive pomace while reducing processing time and costs. While these techniques have been evaluated isolated, the benefits of using both processing strategies simultaneously remain largely unexplored. This study demonstrates the effectiveness of the integration and processing optimization of two environmentally friendly technologies as a promising alternative to treat agro-industrial byproducts.

## 1. Introduction

Olive pomace (OP), the main solid by-product of olive oil production, is composed of olive pulp, pit, and skin. The two-phase centrifugation system for oil extraction generates about 80 kg of OP and 20 kg of oil per 100 kg of fresh olives [1,2]. About 22.3 million tons of olives were produced in 2021, of which nearly 70% was sold to the oil industry, yielding 3.1 million tons of oil, and consequently, 12.5 million tons of OP [3]. The Sustainable Development Goals for 2030 include the efficient use of natural resources and the environmentally sound management of waste. The presence of high-added value compounds, such as dietary fibers, fatty acids, pigments, and phenolic compounds, in the OP has encouraged the study of different extraction processes to recover natural compounds for pharmaceutical, cosmetic, and food applications from this low-cost and abundant material [4,5,6,7]. 

Phenolic compounds, such as hydroxytyrosol, tyrosol, phenolic acids, and flavonoids [8], are amongst the most valuable compounds found in the OP. These compounds have attracted considerable interest due to their potential health-promoting benefits, including antibacterial, anti-inflammatory, antioxidant, and chemopreventive effects [9,10,11]. Conventional extraction of phenolic compounds from agro-industrial by-products typically uses organic flammable solvents and elevated temperatures. Conversely, alternative methods that use mild temperatures and aqueous or “green” solvents, such as high-pressure processing, supercritical fluid extraction, and ultrasound-, microwave-, and enzyme-assisted extractions, have been developed to maximize the extraction of phenolic compounds and reduce the use of harmful solvents. Given that phenolic compounds are bound to cell wall polysaccharides or contained within the various cell structures (vacuoles, chloroplasts, endoplasmic reticulum, and intercellular space) and that high temperature can degrade these compounds, these alternative processes can increase extraction yields by disrupting the cell wall matrix under mild conditions, thereby preserving the phenolic compounds’ functionality [12,13,14]. The effectiveness of enzyme-assisted extraction relies on both the type of enzyme used and the specific reaction conditions applied. Cellulases and pectinases stand out for their ability to hydrolyze cell wall polysaccharides, facilitating the release of phenolic compounds and rendering intracellular materials more accessible for extraction [15]. On the other hand, tannase operates by hydrolyzing ester bonds within carbohydrates and/or proteins, as well as the depsidic linkages between phenolic rings. This specific mode of action not only alters the phenolic profile but also modifies their biological activities [16].

In a previous study with OP, we demonstrated that microwave-assisted aqueous extraction (MAE) achieved similar phenolic extractability to conventional solvent extraction. However, it did so in significantly less time and at higher solids-to-liquid ratio, resulting in reduced water usage and decreased effluent production. In the same study, the integration of MAE and enzyme-assisted extraction (EAE), employing cellulase, pectinase, and tannase, led to increased extractability of phenolic compounds, and faster heating strategy, and yielded extracts with unique phenolic profiles [17]. Nevertheless, the influence of critical extraction parameters, such as temperature, amount and type of enzyme, and the ratio of pomace to solvent, on extraction yield and composition remains to be evaluated. Microwave- and enzyme-assisted extraction (MEAE) holds the potential to significantly enhance the extractability of phenolics from OP if optimum extraction conditions are adequately identified. 

The aim of this study was to optimize the impact of key extraction conditions in the MEAE of phenolic compounds from OP, using tannase alone (MEAE-Tan) or in combination with cellulase and pectinase (MEAE-Tan-Cel-Pec). First, significant factors (temperature, heating ramp and holding time, pomace/water ratio, and enzyme concentration) were identified by a fractional factorial design (FFD) and were subsequently optimized by a central composite rotatable design (CCRD). Response surface methodology (RSM) was conducted to determine the optimal values of each significant factor. Extracts from all experimental conditions were characterized for total phenolic content, antioxidant activity, and phenolic compounds’ profile to determine the optimal conditions to extract OP phenolic compounds with desired yields and composition. The novelty of this study relies on the integration and optimization of the use of green technologies such as microwave- and enzyme-assisted extraction (MEAE) to improve the extraction efficiency of bioactive phenolics from olive pomace while reducing processing time and costs. While these techniques have been evaluated in isolation [18,19], the benefits of using both processing strategies simultaneously remain largely unexplored, with limited knowledge about the impact of key extraction fundamentals on processing efficiency. The primary aim of this study was to elucidate the role of essential extraction parameters in MEAE of phenolics from olive pomace. Additionally, this study aimed to pinpoint the optimal conditions that would effectively enhance the utilization of this promising extraction approach. This study highlights the effectiveness of the integration of two emerging green technologies, under optimized conditions, as a promising alternative to treat agro-industrial byproducts to obtain added-value bioactive compounds. Simultaneously, it reduces both processing time and costs, emphasizing the potential for sustainable and economically viable solutions.

## 2. Materials and Methods

### 2.1. Chemicals and Reagents

Phenolic compound standards, Folin–Ciocalteu phenol reagent, potassium persulfate, Trolox, 2,2-Azino-bis (3-ethylbenzothiazoline-6-sulfonic acid) (ABTS), and 2,2-Diphenyl-1-picrylhydrazyl (DPPH) were purchased from Sigma–Aldrich (>99.5%, St. Louis, MO, USA). Formic acid and acetonitrile were purchased from Fisher Scientific (HPLC grade, 99.9%, Waltham, MA, USA). Cellulase (E.C. 3.2.1.4) and pectinase from *Aspergillus niger* (E.C 3.2.1.15) were purchased from Bio-Cat Inc. (Troy, VA, USA). Tannase (E.C. 3.1.1.20) from *Paecilomyces variotii* was produced at UNICAMP, Brazil [20].

### 2.2. Olive Pomace (OP) Characterization

The OP was provided by a commercial oil mill (Winters, CA, USA). A combination of Biancolilla, Cerasuola, and Nocellara cultivars was processed, and the slurry was centrifuged using a multi-phase decanter (Pieralisi Leopard DMF) to separate the olive oil from the olive pomace. The OP was stored at −20 °C until further use [21]. 

### 2.3. Microwave- and Enzyme-Assisted Extraction (MEAE)

Microwave treatments were performed with the CEM MARS 6^TM^ microwave system (CEM Corporation, Matthews, NC, USA). The microwave system runs at 2455 MHz with a power range between 0 and 1800 W. The system is composed of a multimode microwave generator, a magnetron, and a vessel turntable that allows for the use of up to 24 extraction vessels (20 mL). The system is equipped with a magnetic stirring system to enhance sample homogenization via the addition of stir bars. The temperature of a control vessel, filled with water and OP, was monitored during the extraction using a temperature fiber optic probe inside the control vessel. The variables ramp time, holding time, and final temperature were set for each treatment, while power (W/s, power per seconds) was the result of each experimental design condition, which operates in a continuous mode. After heating, samples were immediately cooled in an ice water bath until 4 °C. Samples were then stored at −80 °C until subsequent analysis.

Because of the large number of experimental variables and responses, a fractional factorial design (FFD) and a central composite design (CCD) were used for MEAE optimization. First, significant variables were identified by the use of an FFD; then, CCD experiments were conducted to determine the optimal value of each significant variable.

Four enzymes were evaluated in the extraction of phenolic compounds from OP with MEAE. These enzymes included commercial cellulase and pectinase, as well as tannase produced by the Bioprocessing Lab (Unicamp, SP, Brazil) utilizing *Paecilomyces variotii* CBMAI 1157. Cellulases and pectinases were selected based on their ability to hydrolyze cell wall polysaccharides, thereby releasing phenolic compounds and rendering intracellular materials more accessible for subsequent extraction. On the other hand, tannase hydrolyzes ester bonds within carbohydrates and/or proteins, and depsidic linkages between phenolic rings, which can alter the phenolic profile and, in this way, also modify their biological activities. These enzymes have been studied and applied by our research group in other studies, with results demonstrating the beneficial effects of these enzymes on the selective extraction of phenolic compounds [17,22].

#### 2.3.1. Evaluation of Reaction Parameters Using a 2^4−1^ Fractional Factorial Design (FFD)

About 1.0 g of OP was weighed in flat-bottomed microwave vessels. Aqueous enzyme solution (tannase; or tannase, pectinase, and cellulase) and water were added to each vessel, and the slurry pH was adjusted to 5.0. Samples were submitted to microwave irradiation for 30 min using different ramp and holding times. The four experimental factors investigated were temperature (X1) from 40 to 60 °C, ramp and holding time (X2) from 5–25 to 25–5 min, pomace/water ratio (X3) from 1:15 to 1:7 (*w*/*w*), and enzyme concentration (X4) from 0 to 1% (*w*/*w*). The range of the experimental factors was selected based on a preliminary study and available literature [17,22,23]. Each parameter was evaluated at three levels (−1, 0, and 1) and varied according to the experimental design presented in Table 1. Extraction parameters were evaluated with respect to total phenolic content of the extracts, which was presented as a mean of two replicates.

#### 2.3.2. Optimization of Microwave- and Enzyme-Assisted Extraction (MEAE) of Phenolic Compounds Using a Central Composite Rotatable Design (CCRD) and Response Surface Methodology (RSM)

A CCRD was used to optimize conditions for phenolic extraction from OP. Two CCRDs were used to evaluate MEAE using tannase alone (MEAE-Tan, Table 2) and MEAE using a combination of tannase with cellulase and pectinase (MEAE-Tan-Cel-Pec, Table 3). Based on the results of the 2^4−1^ FFD, enzyme concentration and pomace/water ratio were selected as independent variables (factors) for the CCRD experiments for MEAE-Tan, while enzyme concentration, temperature, and pomace/water ratio were selected as independent variables for optimization of MEAE-Tan-Cel-Pec. 

Extraction kinetics were evaluated at 5, 15, and 30 min of reaction, and the total phenolic content was the dependent variable (response). For MEAE-Tan, a 2^2^ CCRD was employed with each factor evaluated at five levels (−1.41, −1, 0, 1, and 1.41). The CCRD consisted of 11 experiments, which were used to develop a quadratic model (Table 2). For MEAE-Tan-Cel-Pec, a 2^3^ CCRD was employed, with each factor evaluated at five levels (−1.68, −1, 0, 1, and 1.68). The CCRD consisted of 17 experiments, which were used for building quadratic models (Table 3). The response of each run is the mean of two replicates. Experimental results were analyzed using Statistica 12.0.

### 2.4. Determination of Total Phenolic Content

The total phenolic content of extracts was determined with the Folin–Ciocalteu micro-method [17]. Briefly, 20 μL of the extract solution was mixed with 1.6 mL of distilled water and 100 μL of Folin–Ciocalteu reagent. Then, 300 μL of 20% Na_2_CO_3_ was added after 1 min and again after 8 min. Subsequently, the mixture was incubated in a shaking incubator at 40 °C for 30 min, and absorbance was measured at 765 nm. A calibration curve was constructed using gallic acid as a standard (25–500 mg/mL). The absorbance of the extracts was measured within the limits of the calibration curve. Distilled water was used for background subtraction. Measurements were performed in triplicate, and results are presented as mean. Total phenolic content was calculated according to the linear equation of the calibration curve and expressed as mg of gallic acid equivalents (GAE)/kg pomace (dry weight).

### 2.5. Antioxidant Capacity Evaluation

#### 2.5.1. DPPH Assay

DPPH radical scavenging activity was evaluated by determining DPPH at a steady state in methanol solution after adding the mixture of antioxidants [24]. Extracts were dissolved in 10 mL of absolute methanol to produce a final concentration of 4 mg/mL. Then, 2 mL of 0.004% DPPH (0.2 mM) in methanol was added to 1 mL of the extract solution. After shaking the mixture vigorously, the decrease in absorbance was measured at 517 nm until the reaction reached a plateau. Absolute methanol was used as the control, and Trolox, a stable antioxidant, was used as standard. DPPH radical-scavenging activity is expressed in mmol Trolox equivalents (TE)/g extract. Analyses were carried out in triplicates.

#### 2.5.2. ABTS Assay

The ABTS radical was produced by a reaction of 7 mM ABTS with 2.45 mM potassium persulfate in the dark at room temperature (25 °C). The mixture was allowed to stand for 16 h before use. The aqueous ABTS^+^ solution was diluted with water to obtain an absorbance of 0.7 at 734 nm. Subsequently, 3 mL of ABTS^+^ solution was added to 30 μL of the extracts [25]. After 6 min, absorbance was measured at 734 nm using a Thermo Genesys 10S spectrophotometer (Thermo Fisher Scientific, Waltham, MA, USA). Trolox was used as a standard. Analyses were performed in triplicates, and the results were expressed as mmol TE/g extract.

### 2.6. Identification and Quantification of Phenolic Compounds Using HPLC-DAD

The phenolic profile of OP extracts obtained with MEAE under non-optimized conditions, after optimization, and by enzyme-assisted extraction process (EAEP) were determined according to the procedures of Crawford et al. [26]. EAEP was carried out in a thermostatic water bath for 60 min, without microwave irradiation. Extracts were diluted with water (1:2) before injection. Analysis was performed on an Agilent 1290 Infinity II LC system equipped with a diode-array detector (DAD) (Agilent Technologies, Santa Clara, CA, USA) and an Agilent C18 Eclipse Plus column (5 μm, 4.6 × 250 mm) at a flow rate of 1.2 mL/min. The mobile phase consisted of 3.0% acetic acid in water (A) and methanol (B). The used elution gradient was as follows: 5% B at 0 min; 12% B at 2 min; 22% B at 10 min; 36% B at 15 min; 50% B at 35 min; 100% B at 37 min; 100% B at 38 min; 5% B at 40 min; and 5% B at 45 min. The injection volume was 20 μL. DAD signals were recorded at 280 nm, 320 nm, and 365 nm. Stock solutions of individual compounds were prepared in methanol and used to prepare standard mixtures in methanol with the following concentrations: 1000 μg/mL oleuropein; 400 μg/mL hydroxytyrosol, verbascoside, rutin, and luteolin-7-glucoside; 200 μg/mL tyrosol and quercetin; and 80 μg/mL caffeic acid. A six-point linear calibration curve in 1:1 (*v*/*v*) methanol/water was constructed using standards stored at −20 °C.

### 2.7. Statistical Analysis

Experimental results of the 2^4−1^ FFD and CCRD were analyzed using Statistica 12.0, and the TPC measurements were presented as the mean of triplicate measurements. Experimental results obtained using HPLC are expressed as mean ± SD. Statistical difference between treatments for polyphenolic quantification was assessed using an unpaired *t*-test (*p* ≤ 0.05). Statistical analysis was performed using the software Minitab version 16.1.1.

## 3. Results and Discussion

### 3.1. Evaluation of Extraction Parameters Using a 2^4−1^ FFD

OP was composed of 6.4% lipids, 2.8% proteins, 15.8% total carbohydrates, and 75% moisture, in accordance with the values reported in the literature [27]. 

Temperature, microwave ramp time/holding time, pomace/water ratio, and enzyme concentration were evaluated for the extractability of phenolic compounds from OP using tannase alone or a combination of tannase, cellulase, and pectinase under microwave irradiation (MEAE-Tan and MEAE-Tan-Cel-Pec, respectively), as described in Table 4 and Table 5. The aim of these experiments was to determine the significant variables and subsequently their optimum levels in the final optimization step using the CCRD.

Extraction temperatures were applied using microwave heating and varied from 40 to 60 °C. The temperature range was defined based on the optimum conditions for each enzyme, aiming for their maximum activity and stability. Pomace/water ratios were chosen taking into account the results of our preliminary studies, aiming to reduce the amount of water used during processing to minimize waste production and facilitate downstream steps for the concentration of bioactive compounds [17]. To determine the significance of enzyme concentration on phenolic extractability, enzyme concentration varied from 0 to 1%. To the best of our knowledge, this is the first study to perform a comprehensive processing optimization to elucidate the impact using tannase alone or a combination of other enzymes under microwave irradiation for the extraction of olive pomace’s phenolic compounds, which brings relevant knowledge for the application of this new technology since not all enzymes are stable under microwave irradiation.

Table 4 describes the statistical significance of the variables evaluated on phenolic extractability (FFD). Total phenolic extractability with MEAE-Tan ranged from 176 to 473 mg GAE/kg, and the highest phenolic contents were achieved in runs 2 and 3 (Table 4). All extractions lasted 30 min, which shows a productivity gain when using microwave irradiation in comparison with conventional extraction, as we previously investigated [17]. Statistical analysis revealed that, for the levels evaluated, the only statistically significant variable for total phenolic content was enzyme concentration (Appendix A). Tannase concentration had a positive effect on total phenolic extraction. Although the other variables were not statistically different within the range evaluated, they presented different trends. High temperatures showed a positive effect on phenolic extraction, which was expected, since tannase’s optimal temperature for activity is 60 °C [20]. Also, higher heating ramp times and shorter hold times seemed to have a beneficial effect on phenolic extraction. The pomace/water ratio, although not significant, had a negative effect on the process, suggesting that high water concentrations had a positive effect on the extraction, in agreement with the literature findings.

When looking at the synergistic impact of the different enzymes used, phenolic extractability with MEAE-Tan-Cel-Pec ranged from 163.67 to 382.88 mg GAE/kg (Table 5), being lower than that obtained using MEAE-Tan, i.e., 176.20–473.31 mg GAE/kg (Table 4). The higher extractability observed for MEAE-Tan highlights the effectiveness of the mode of action of tannase, which hydrolyzes ester and depsidic linkages between phenolics. However, the presence of other enzymes in semi-purified tannase extracts may have also contributed to the release of phenolics from the cell matrix, which may explain the higher MEAE-Tan phenolic extraction. Higher phenolic contents of MEAE-Tan-Cel-Pec extracts were observed in runs 2 and 8 (382.88 and 377.72 mg GAE/kg, respectively). Temperature, pomace/water ratio, and enzyme concentration significantly affected OP phenolic extraction with MEAE, as shown by the results of the statistical analysis (Appendix A). As shown in Table 5, lower TPC values were observed at the center points (runs 9, 10, and 11), being lower than those obtained in runs 1, 4, 6, and 7, in which enzymes were not employed (only microwave irradiation). This may have occurred because, in the runs carried out without enzymes, either the temperature was higher than that applied at the center points (>50 °C), the pomace/water ratio was lower than the ones used in the central points (<1:10), or the microwave power was higher than that applied at the center points. This shows that temperature and heating profile (ramp and holding time) influenced the extraction process. Higher phenolic extractability at higher temperature could be due to increased solubility of these compounds in the reaction medium [28]. Appendix A shows that although the variable ramp and holding time was not significant, its effect was negative, which means that the heating profile of 5 min ramp and 25 min of holding time was more efficient than 25 min ramp time with 5 min of holding time, as observed in runs 4 and 6.

Temperature had a strong positive effect on total phenolic content obtained with MEAE-Tan-Cel-Pec, indicating that higher temperatures had a positive effect on the extractability of phenolics, regardless of the use of enzyme or not (Appendix A). Because enzymatic extractions may lead to higher phenolic extractability, its use could be further exploited in conjunction with microwave processing to ensure higher phenolic compounds’ release at reduced processing time. This pattern was opposite to the one observed for MEAE-Tan, due to the characteristics of each enzyme. Pectinase and cellulase are stable and more active at temperatures below 60 °C. The addition of tannase to pectinase and cellulase seemed to increase the optimum temperature activity to values greater than 50 °C. 

The effect of enzyme concentration on total phenolic content obtained with MEAE-Tan-Cel-Pec was also positive, indicating that concentrations above zero favored phenolic extraction (Table 5). A positive and significant effect was observed for the pomace/water ratio. Although the effect of pomace/water ratio was not as high as that of temperature and enzyme concentration, these results indicate that higher yields can be achieved using less water (i.e., using a more concentrated slurry). Reduced water usage for the MEAE-Tan-Cel-Pec represents an advantage over the MEAE-Tan process. Ramp and holding time were the only non-significant variable. Although its effect was not significant, ramp time presented a negative effect, which meant that shorter ramp times and longer holding times might favor phenolic extraction. This heating profile uses high power values for a very short time, which means that much of the process is carried out at low power, probably favoring enzyme stability. High microwave power can increase temperature excessively, leading to enzyme and protein denaturation and loss of bioactivity [29]. Jawad and Langrish tested three different power levels (400, 600, and 800 W) to extract phenolic acids from mandarin peels [28]. The authors concluded that the final concentration of phenolic acids was not affected by higher microwave power, but, as microwave power increased, extracts became darker, suggesting the formation of Maillard reaction products [30].

Based on the FFD results, enzyme concentration and pomace/water ratio were selected for subsequent optimization of the MEAE-Tan process, while enzyme concentration, pomace/water ratio, and temperature were selected for the optimization of the MEAE-Tan-Cel-Pec process using a CCRD. 

### 3.2. Phenolic Extraction Optimization of MEAE-Tan Using a 2^2^ CCRD and Extract Characterization

CCRD experiments were conducted at a maximum temperature of 60 °C, and extraction kinetics were performed at each experimental condition to investigate the role of microwave exposure time (5, 15, and 30 min). For MEAE-Tan-Cel-Pec, a CCRD was carried out with three independent variables: temperature, enzyme concentration, and pomace/water ratio. As performed for the evaluation of the effects of the variables evaluated in the MEAE-Tan experiments (Table 5), runs were conducted using 5 min of ramp time and 0, 15, or 25 min of holding time to achieve reaction times of 5, 15, or 30 min. Phenolic extractability for all CCRD extraction experiments for MEAE-Tan are presented in Table 6. According to Table 6, overall higher phenolic extractability was achieved in 15 min of reaction, regardless of the experimental conditions evaluated, which could be explained by the phenomenon of “hydration” that takes place in the first minutes of extraction (0–5 min), followed by subsequent diffusion of phenolic compounds into the water, which may have been enhanced by increasing microwave energy from 58 to 287 W, until temperature reached 60 °C. Under localized heating, pressure builds up within cells, rupturing cell walls and increasing extraction efficiency and speed. After 5 min of reaction, phenolic extraction continued to increase, with maximum phenolic recovery at 15 min of reaction. Higher extractability at 15 min can be attributed to the ability of tannase to hydrolyze the cross-links between the cell wall and phenolics, thus facilitating solvent penetration into the cells and release of phenolics from the vacuoles into the water. After the hydration effect and enzyme action, a rapid increase in diffusion occurred in the first 15 min of reaction, yielding the highest phenolic yields. After 30 min of reaction, a reduction in total phenolic content was observed, which may be due to heat exposure, oxidation reactions, or degradation of the phenolics. In addition, rearrangement of phenolic compounds may have occurred, as certain compounds were not detected with the analytical method [28,31,32]. Because of the higher phenolic extractability observed at 15 min, TPC values at 15 min were used to develop the mathematical model for MEAE-Tan.

Optimum extraction conditions for phenolics were determined by multiple regression analysis of the data. The estimated regression model (Equation (1)), correlating the recovery of phenolic compounds with MEAE-Tan variables, is given below in terms of significant factors, where *Y* is the total phenolic content (TPC), *X*_1_ is tannase concentration (*w*/*w*), and *X*_2_ is pomace/water ratio (*w*/*w*).
(1)$$Y=2929.1+970.57X_{1}+10.84X_{1}^{2}-1193.4X_{2}+615.09X_{2}^{2}$$

The estimated effect for each variable, as well as the interactions between them, were determined and are presented in Appendix A. The two variables, pomace/water ratio and enzyme concentration, were statistically significant at the 95% confidence level, although the interaction between *X*_1_ versus *X*_2_ was not statistically significant. Appendix A shows the analysis of variance (ANOVA) of total phenolic content. The pure error was very low, indicating good reproducibility of the experimental data, and the correlation coefficient (0.94) indicates that the model (Equation (1)) represents well the relationship between response and variables. Lack of fit test was performed by comparing the ratio of the mean squared lack of fit and the mean squared pure error at the 95% confidence level. The calculated F_3,2_ value (12.93) was lower than the tabled F_3,2_ value (19.16), indicating that the lack of fit of the model was not significant and that the model is reproducible. In addition, we calculated the coefficient of determination (R^2^) and coefficient of variation (CV) and conducted a correlation analysis between predicted and experimental values (Appendix A). The R^2^ value was 0.94, indicating that 94% of variations could be explained by the fitted model. The relatively low CV value (5.0%) indicates that the model is reproducible. 

The optimal conditions for MEAE-Tan extraction of phenolic compounds from OP using tannase were determined using RSM, presented in Figure 1. According to the RSM, the optimal conditions for phenolic extraction with MEAE-Tan was 60 °C, 15 min of reaction, 2.34% of tannase (*w*/*w*), and 1:15 (g/mL) pomace/water ratio. The suitability of the model equation for predicting the optimum response was tested using these theoretical optimal conditions. The predicted extraction of phenolics was 7110.6 mg GAE/kg, which was in good agreement with the experimental value of 7583.11 mg GAE/kg (6.6% above the predicted). The strong correlation between experimental and predicted results confirmed that the response model was adequate for reflecting the expected optimization.

As presented in Figure 1, extraction of phenolic compounds increases with increasing water and enzyme concentrations, within the ranges evaluated. These conditions were tested experimentally, and the validation of the proposed model is shown in Appendix A. The optimized reaction resulted in the production of extracts with 7583.11 mg GAE/kg of pomace in 15 min, a concentration 16 times greater than the maximum concentration obtained before optimization (473.31 mg GAE/kg of pomace, Table 6).

A previous study with OP, comparing conventional (methanol 80%) to pressurized liquid extraction (PLE), showed total phenolic contents ranging from 281 to 1659 mg of GAE/kg of dried OP [6]. In another study investigating the impact of microwave-assisted extraction parameters using dried OP, a higher total phenolic content (985 mg/kg) was achieved compared to the conventional industrial solvent extraction method [33]. These data, from previous literature, highlight the effectiveness of the integration of two emerging technologies (microwave- and enzyme-assisted extractions), which greatly improved the total phenolic content of the extracts obtained after optimization of MEAE-Tan extraction of OP.

Because phenolic compounds in plants are linked to the cell wall or are contained in vacuoles, several factors can affect the release of these compounds from these cell structures, including temperature, solvent, microwave irradiation, and enzyme concentration in the reaction medium [15,34,35]. Enzyme concentration is a critical variable in enzymatic extraction processes. Enzymes have been previously applied to improve the extraction of phenolic compounds, and hydrolytic enzymes such as tannase have been shown to release phenols from the cell wall matrix by degrading polysaccharides and breaking ester linkages between phenols and cell wall polymers [36,37,38]. However, the amount of enzyme applied for extraction must be evaluated to avoid excessive use, which would unnecessarily increase extraction costs.

The pomace/water ratio significantly affected phenolic extractability. Figure 1 showed that the total phenolic content of OP extracts increased significantly when the pomace/water ratio was reduced from 1:4 to 1:15. A similar effect was reported in the extraction of phenolic compounds from other plant sources in previous studies [39,40]. Increasing water concentration enhances gradient concentration between the extraction medium phenolics and the matrix phenolics, thus favoring the extractability and diffusion of phenolics into the extraction medium [41]. In addition, increased water concentration leads to reduced solids in the reaction medium, which would reduce its viscosity and favor overall mass transfer. Temperature is one of the key variables affecting the release of phenols from vegetable matrices, as it modifies equilibrium and mass transfer conditions in solid–liquid extractions and affects enzyme activity [28]. However, high temperatures should not be increased indefinitely because enzyme denaturation and phenolic degradation may occur. For MEAE-Tan, maximum temperature of 60 °C was considered as the optimum temperature for tannase activity and stability and phenolic extraction.

The extracts obtained in MEAE-Tan optimization experiments (CCRD) were analyzed for their antioxidant capacity (Figure 2) and their phenolic profiles were characterized (Table 7). The antioxidant capacity of the extracts is related to the amount and type of antioxidant molecules they contain. Depending on the extraction conditions used (temperature, enzymes, pomace/water ratio, and microwave irradiation), extracts with different antioxidant activity, total phenolic content, and phenolic profile were obtained. As presented in Figure 2, runs 6 and 7 produced extracts with the highest antioxidant capacity, 59.82 and 74.54 mmol TE/g, respectively. Run 5, which had the lowest enzyme and water concentrations, resulted in the production of an extract with the lowest antioxidant capacity in DPPH and ABTS assays. Antioxidant activity varied in accordance with the amount of total phenolic compounds extracted, which can indicate the effective release of bioactive phenolics with tannase [22,23,42]. These data agree with those reported by Kessy et al. (2018). The authors showed that aqueous enzymatic treatments (pectinase, tannase, and β-glucosidase) enhanced the release of phenolic compounds from lychee pericarp extract and their antioxidant activity by increasing the release and biotransformation of phenolic compounds [38]. The correlation between total phenolic content measured by the Folin–Ciocalteu method and the antioxidant capacity of coffee, tea, beer, apple juice, and other phenolic extracts has been observed by other authors [43].

Table 7 presents the phenolic profile of extracts obtained by different extractions: MEAE-Tan under non-optimized conditions and MEAE-Tan after optimization. The major phenolic compounds found in OP were identified using HPLC-DAD.

Optimized MEAE-Tan promoted a marked increase in the extraction of chlorogenic acid, which was present at a concentration ~40 times greater than the one obtained by MEAE-Tan under non-optimized conditions. Also, chlorogenic acid was the major phenolic compound found in the MEAE-Tan extract, at concentrations of ~10 times greater than the other quantified phenolic compounds. Our results demonstrate that tannase-assisted extraction enhances the extraction of chlorogenic acid, a phenolic compound that exerts beneficial effects on health [44,45,46]. *P. variotii* tannase has been shown to possess β-glucosidase and esterase activities [16]. These activities result in phenolic compounds with low molecular weight, increasing the concentration of hydroxytyrosol and chlorogenic, hydroxyphenylacetic, caffeic, ferulic, and *m-*coumaric acids.

### 3.3. Phenolic Extraction Optimization of MEAE-Tan-Cel-Pec Using a 2^3^ CCRD and Extract Characterization

The total phenolic content of the extracts obtained with the 2^3^ CCRD experiments for MEAE-Tan-Cel-Pec are presented in Table 8. According to Table 8, 12 runs presented similar or slightly higher amounts of phenolic compounds at 15 min and 30 min of reaction, indicating that phenolic concentration would not increase after 15 min of extraction, under the studied conditions. Also, after 15 min, phenolic yield started to decrease in some runs. Prolonged exposure to microwaves irradiation may lead to degradation of phenolic compounds and denaturation of enzymes. Previous study evaluated phenolic extraction from mandarin peels using microwave energy. Extractions were carried out using water as solvent for periods of 1 min, 3 min, and 12 min at 135 °C. The results showed that the maximum phenolic yield was obtained after 3 min of reaction and, with an increase in extraction time, the extent of antioxidant degradation increases [30]. Similar results were obtained for flavanones extraction under 10 min of microwave irradiation, which decreased the extraction yield because of the low solubilization of flavanones in the system, causing thermal degradation of flavanones [47]. These results indicate that the combination of time and temperature is crucial for phenolic compounds’ extraction.

Based on Table 8 results and aiming to maximize phenolic extraction from OP, we analyzed the data obtained at 15 min of reaction using the validated mathematical model to predict the TPC yields for MEAE-Tan-Cel-Pec extraction (Appendix A). The estimated effect of each variable and interactions between variables were determined and are shown in Appendix A. Enzyme concentration and pomace/water ratio were significant at 95% confidence level. Appendix A shows the total phenolic content in the extracts obtained in the experiments and the predicted values according to the second-order regression model. Equation (2) correlates the recovery of phenolic compounds obtained with MEAE-Tan-Cel-Pec, where *Y* is the total phenolic content; *X*_1_ is the temperature, *X*_2_ is enzyme concentration, and *X*_3_ is the pomace/water ratio (*w*/*w*).
(2)$$Y=696.33-55.95X_{1}+72.73X_{1}^{2}+153.12X_{2}+31.25X_{2}^{2}-458.12X_{3}+327.40X_{3}^{2}\;$$

Appendix A shows the analysis of variance (ANOVA) of the responses (total phenolic content) of MEAE-Tan-Cel-Pec runs at 15 min of reaction. The pure error was 6.4, which was considered low, indicating good reproducibility of data. The correlation coefficient (0.93) and the F-test result (F_cal_ is 3.07 times higher than F_tab_) indicate that the mathematical model (Equation (2)) represents well the relationship between response and variables. We also determined the coefficient of determination (R^2^) and coefficient of variation (CV) of the model and performed a correlation analysis between predicted values and experimental values (Appendix A). R^2^ was 0.93, indicating that variations were well explained by the model. The low CV value (6.4%) suggests that the model is reproducible. The optimal conditions for phenolics extraction from OP with MEAE-Tan-Cel were 46 °C, 2.5% of enzymes (*w*/*w*), and 1:20 (g/mL) pomace/water ratio. Under these conditions, phenolic extraction increased from 382 mg GAE/kg to 2938.25 mg GAE/kg, a 7.7-fold increase compared with non-optimized extraction conditions. The suitability of the mathematical model for predicting the optimum response values was tested using the optimal conditions. The predicted extraction of total phenolic was 2793 mg GAE/kg, which was consistent with the experimental value of 2938.25 mg GAE/kg (5% above the predicted). The strong correlation between experimental and predicted results confirmed that the response model was adequate to predict the optimized results (Appendix A).

Figure 3 shows extraction conditions leading to optimum phenolic extractability using MEAE-Tan-Cel-Pec. In Figure 3a, the concentration of phenolics increases at extreme temperature points (46 °C and 73 °C), presenting a valley at 60 °C. Regarding enzyme concentration, 2.5% enzyme was the optimum theoretical value for phenolic extraction (Figure 3b) using the highest pomace/water ratio of 1:20 (Figure 3c).

Previous studies observed that sample-to-solvent ratio had a significant effect on total phenolic extraction from OP. The optimum conditions determined by the authors using RSM were 1:60 (g/mL) pomace/solvent ratio, 90 °C, and 70 min of reaction [48]. This is consistent with mass transfer principles, which define that the concentration gradient (the driving force) is higher when there is more solvent present, leading to higher diffusion rates. Also, Figure 3c shows that, within the range of enzyme concentration evaluated (0.5–2.5%), phenolic extraction is high when a low pomace/water ratio is used. This finding enables the use of low enzyme concentrations, which can reduce processing costs.

The extracts obtained in MEAE-Tan-Cel-Pec optimization experiments were analyzed for antioxidant capacity and phenolic profile. These data are presented in Figure 4 and Table 9, respectively. 

Antioxidant activity of OP extracts obtained with MEAE-Tan-Cel-Pec ranged from 10 to 35 mmol TE/g and was high in extracts with high phenolic concentrations (runs 3, 4, and 13), as observed compared to MEAE-Tan. These results indicate that the extracted phenolic compounds are mainly responsible for the scavenging activity of OP extracts and that tannase, cellulase, and pectinase were effective in releasing bioactive phenolics from OP. 

The use of different enzymes (Celluclast 1.5 L, Pectinex Ultra, and Novoferm) to release phenolics from grape wastes has been shown to influence the phenolic content and antioxidant capacity of grape extracts. In the aforementioned study, a good correlation between antioxidant activity and phenolic content was observed [49]. However, prolonged exposure to enzymes may reduce antioxidant activity of bound and free polyphenols [50].

The phenolic profile of extracts obtained with MEAE-Tan-Cel-Pec before and after optimization are presented in Table 9. Similar to MEAE-Tan, the optimum conditions for MEAE-Tan-Cel-Pec promoted a marked increase in the extraction of chlorogenic and ferulic acids. We did not observe a significant increase in vanillic and *m-*coumaric acids, which may have resulted from potential degradation of these phenolics into compounds of lower molecular weight. 

Although the optimized treatment with tannase (MEAE-Tan) released higher concentrations of chlorogenic acid compared with the MEAE-Tan-Cel-Pec, the optimized treatment with the mixture of enzymes (MEAE-Tan-Cel-Pec) increased the release of several other phenolics, such as hydroxytyrosol and hydroxyphenylacetic, caffeic, ferulic, and elenolic acids. Based on the above-described results, the integration of microwave processing and the three evaluated enzymes (Tan-Cel-Pec) represents a promising alternative strategy to enhance the overall extractability of OP phenolics. The enzymes acted in synergy under microwave conditions and, although the biological activity and stability of the extracts have yet to be investigated, it is possible to visualize the technical and economic potentials that MEAE processes may exhibit in the near future. 

## 4. Conclusions

This study elucidated the role of key extraction conditions (amount and type of enzyme, heating strategies, and pomace/solvent ratio) on the effectiveness of a sustainable integrated approach based on microwave processing and use of selected enzymes with respect to the extractability of OP phenolics. The combination of MEAE processing and tannase alone enabled the release of higher concentrations of total phenolic compounds and chlorogenic acid than MEAE-Tan-Cel-Pec. Conversely, optimum extraction conditions for MEAE-Tan-Cel-Pec favored the release of several other phenolics, such as hydroxytyrosol and hydroxyphenylacetic, caffeic, ferulic, and elenolic acids. Although the biological activity and stability of extracts remain to be evaluated, this study offers compelling evidence of the potential benefits associated with combining two environmentally friendly technologies, microwave processing and enzyme-assisted extraction, to treat inexpensive raw materials, such as OP. This integrated approach resulted in the production of extracts with higher phenolic concentration in shorter incubation times and at reduced temperatures and could therefore be exploited as an economically feasible alternative to conventional extraction methods. Further studies should explore the application of these OP extracts in the field of food and pharmaceutical applications. Moreover, exploring the application of MEAE in other raw materials, such as grape and citrus agro-industrial byproducts, is also warranted.

## Figures and Tables

**Figure 1 foods-12-03754-f001:**
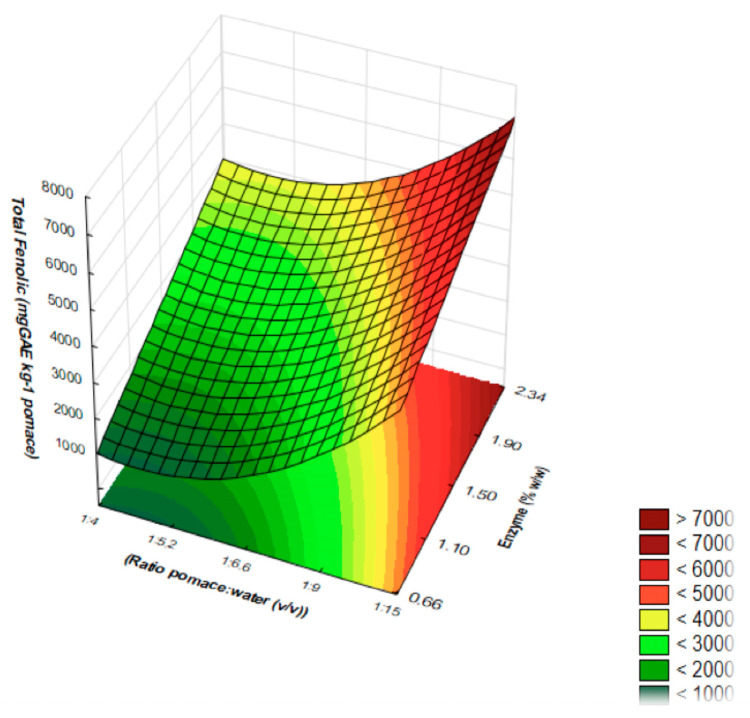
Response surface plot of MEAE-Tan showing the effects of enzyme concentration (*X*_1_) and pomace/water ratio (*X*_2_) on phenolic compounds’ extraction from OP.

**Figure 2 foods-12-03754-f002:**
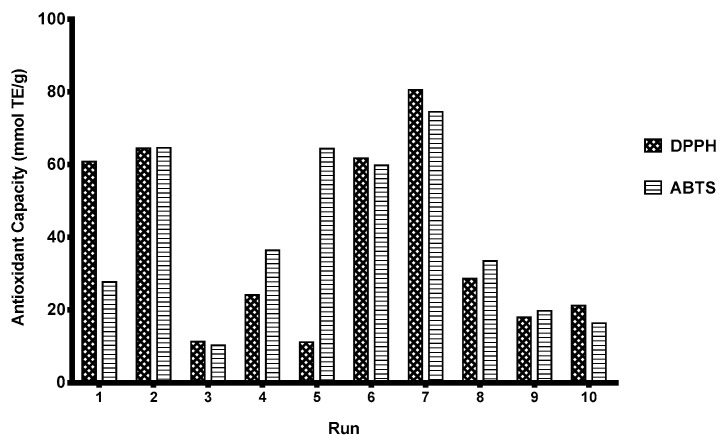
Antioxidant capacity of OP extracts obtained with optimized MEAE-Tan conditions at 15 min using a microwave power of 54–287 W.

**Figure 3 foods-12-03754-f003:**
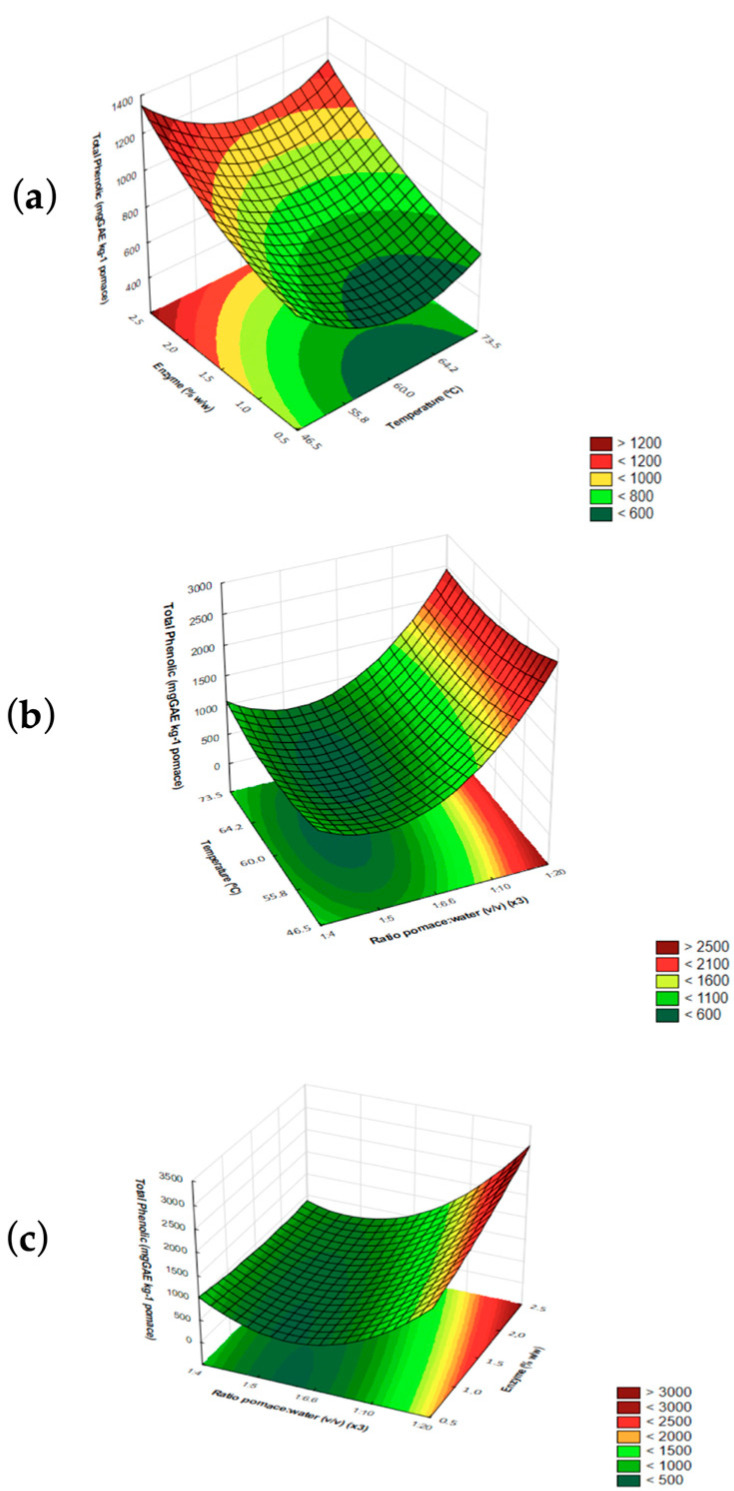
Effect of (**a**) enzyme concentration (*X*_2_) and temperature (*X*_1_), (**b**) pomace/water ratio (*X*_3_) and temperature (*X*_1_), and (**c**) pomace/water ratio (*X*_3_) and enzyme concentration (*X*_2_) on phenolic extractability using MEAE-Tan-Cel-Pec.

**Figure 4 foods-12-03754-f004:**
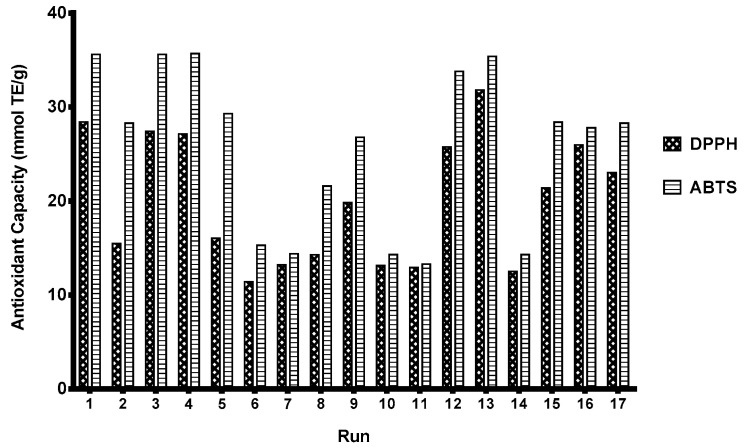
Antioxidant capacity of OP extracts obtained with optimized MEAE-Tan-Cel-Pec conditions (15 min of reaction using a microwave power of 54–287 W).

**Table 1 foods-12-03754-t001:** Factors evaluated in the 2^4−1^ FFD to enhance phenolic extractability from olive pomace with MEAE-Tan and MEAE-Tan-Cel-Pec.

Factors	Levels		
	−1	0	1
X1—Temperature (°C)	40	50	60
X2—Ramp and hold time (min)	5, 25	15, 15	25, 5
X3—Pomace/water ratio (*w*:*w*)	1:15	1:10	1:7
X4—Enzyme concentration (%)	0	0.5	1.0

**Table 2 foods-12-03754-t002:** Factors evaluated in the CCRD for the optimization of MEAE-Tan.

Factors	Levels				
	−1.41	−1	0	+1	+1.41
X1—Enzyme concentration (%)	0.66	0.9	1.5	2.1	2.34
X2—Pomace/water ratio (*w*:*w*)	1:15	1:11	1:6.6	1:4.7	1:4

**Table 3 foods-12-03754-t003:** Factors evaluated in the CCRD for the optimization of MEAE-Tan-Cel-Pec.

Factors	Levels				
	−1.68	−1	0	+1	+1.68
X1—Temperature (°C)	46.5	55	60	65	73.5
X2—Enzyme concentration (%)	0.5	0.9	1.5	2.1	2.5
X3—Pomace/water ratio (*w*:*w*)	1:20	1:11	1:6.6	1:4.7	1:4

**Table 4 foods-12-03754-t004:** The role of the 2^4−1^ FFD experimental factors on phenolic extractability (TPC) from olive pomace with MEAE-Tan. Microwave power range: 70–175 W.

Run	Extraction Conditions (Experimental Factors)				Response
	X1 Temperature (°C)	X2 Microwave Ramp Time, Hold Time (min)	X3 Pomace/Water Ratio (*w*/*w*)	X4 Enzyme Concentration (%*w*/*w*)	Total Phenolic Content (mg GAE ^1^/kg Pomace)
1	40	5, 25	1:15	0	187.13
2	60	5, 25	1:15	1	473.31
3	40	25, 5	1:15	1	461.48
4	60	25, 5	1:15	0	176.20
5	40	5, 25	1:7	1	311.12
6	60	5, 25	1:7	0	219.50
7	40	25, 5	1:7	0	207.99
8	60	25, 5	1:7	1	420.91
9	50	15, 15	1:10	0.5	205.09
10	50	15, 15	1:10	0.5	204.64
11	50	15, 15	1:10	0.5	203.43

^1^ GAE: gallic acid equivalents.

**Table 5 foods-12-03754-t005:** The role of the 2^4−1^ FFD experimental factors on phenolic extractability (TPC) from olive pomace with MEAE-Tan-Cel-Pec. Microwave power range: 70–175 W.

Run	Extraction Conditions (Experimental Factors)				Response
	X1 Temperature (°C)	X2 Microwave Ramp Time, Hold Time (min)	X3 Pomace/Water Ratio (*w*/*w*)	X4 Enzyme Concentration (%*w*/*w*)	Total Phenolic Content (mg GAE ^1^/kg Pomace)
1	40	5, 25	1:15	0	197.45
2	60	5, 25	1:15	1	382.88
3	40	25, 5	1:15	1	232.04
4	60	25, 5	1:15	0	221.12
5	40	5, 25	1:7	1	316.37
6	60	5, 25	1:7	0	260.06
7	40	25, 5	1:7	0	202.54
8	60	25, 5	1:7	1	377.72
9	50	15, 15	1:10	0.5	172.32
10	50	15, 15	1:10	0.5	163.97
11	50	15, 15	1:10	0.5	163.67

^1^ GAE: gallic acid equivalents.

**Table 6 foods-12-03754-t006:** Role of enzyme concentration and pomace/water ratio on the extractability of OP phenolics using MEAE-Tan (2^2^ CCRD and microwave power of 54–287 W).

Run	Extraction Conditions		Total Phenolic Content (mg GAE ^1^/kg Pomace)		
	*X*_1_ Enzyme Concentration (%)	*X*_2_ Pomace/Water Ratio (*w*/*w*)	Reaction Time (min)		
			5	15	30
1	0.9	1:11	1203.77	3611.30	1719.66
2	2.1	1:11	1742.71	5053.86	2526.93
3	0.9	1:4.7	589.40	1768.19	842.00
4	2.1	1:4.7	1186.32	3440.32	1678.20
5	0.66	1:6.6	458.05	1397.06	649.67
6	2.34	1:6.6	1559.49	4678.48	2227.85
7	1.5	1:15	2129.60	6388.80	3071.54
8	1.5	1:4	696.46	2089.38	994.94
9	1.5	1:6.6	981.65	2723.54	1361.77
10	1.5	1:6.6	1006.69	3020.08	1589.52
11	1.5	1:6.6	1019.27	3041.60	1448.38

^1^ GAE: gallic acid equivalents.

**Table 7 foods-12-03754-t007:** Major phenolic compounds identified using HPLC-DAD in extracts obtained with MEAE-Tan before and after optimization.

Phenolic Compound	MEAE-Tan before Optimization * (mg/kg Pomace)	MEAE-Tan after Optimization ** (mg/kg Pomace)
Hydroxytyrosol	21.60 ± 14.05 ^a^	31.27 ± 3.54 ^a^
Protocatechuic acid	14.85 ± 3.45 ^a^	17.45 ± 0.64 ^a^
Chlorogenic acid	19.35 ± 11.48 ^b^	832.06 ± 99.13 ^a^
Tyrosol	4.80 ± 0.15 ^a^	4.81 ± 3.38 ^a^
Hydroxyphenylacetic	-	38.65 ± 5.14 ^a^
acid		
Vanillic acid	34.20 ± 0.20 ^a^	25.20 ± 3.06 ^a^
Caffeic acid	7.65 ± 1.54 ^a^	14.60 ± 1.35 ^a^
Vanillin	3.45 ± 0.42 ^a^	-
*P-*Coumaric acid	7.95 ± 1.45 ^a^	2.66 ± 0.63 ^a^
Ferulic acid	7.95 ± 1.64 ^b^	36.46 ± 6.22 ^a^
*m-*Coumaric acid	0.45 ± 0.19 ^b^	8.11 ± 1.00 ^a^
Elenolic acid	-	-

Values are presented as mean ± standard deviation. ^a,b^ Means within a row followed by different letters differ significantly (*p* ≤ 0.05) using Student’s unpaired *t*-test. * Microwave-assisted enzymatic extraction: 1% (*w*/*w*) tannase, 60 °C, 30 min of reaction under microwave irradiation, and 1:15 pomace/water ratio. ** Microwave-assisted enzymatic extraction under optimized conditions: 2.34% (*w*/*w*) tannase, 60 °C, 15 min of reaction under microwave irradiation, and 1:15 pomace/water ratio. -: not detected.

**Table 8 foods-12-03754-t008:** Impact of temperature, enzyme concentration and pomace/water ratio on the phenolic content of MEAE-Tan-Cel-Pec extracts (microwave power of 25–180 W)**.**

Run	Extraction Conditions			Total Phenolic Content (mg GAE ^1^/kg Pomace)		
	*X*_1_ Temperature (°C)	*X*_2_ Enzyme Concentration ^2^ (%)	*X*_3_ Pomace/Water Ratio (*w*/*w*)	Reaction Time (min)		
				5	15	30
1	55	0.9	1:11	1022.18	1201.58	1301.14
2	65	0.9	1:11	858.04	1037.43	932.49
3	55	2.1	1:11	1515.51	1874.30	1398.01
4	65	2.1	1:11	1412.36	1591.75	1723.61
5	55	0.9	1:4.7	651.76	720.30	945.34
6	65	0.9	1:4.7	491.06	567.22	676.89
7	55	2.1	1:4.7	727.53	803.69	848.24
8	65	2.1	1:4.7	693.26	769.42	825.39
9	46.7	1.5	1:6.6	913.98	1020.90	985.62
10	73.5	1.5	1:6.6	836.99	946.92	889.39
11	60	0.5	1:6.6	574.50	693.72	672.87
12	60	2.0	1:6.6	913.98	1036.94	1136.38
13	60	1.5	1:20	2384.03	2715.01	2501.89
14	60	1.5	1:4	622.44	687.34	547.84
15	60	1.5	1:6.6	522.64	629.56	879.76
16	60	1.5	1:6.6	626.89	735.42	903.82
17	60	1.5	1:6.6	588.40	695.32	874.95

^1^ GAE: gallic acid equivalents; ^2^ enzymes’ final concentration equally distributed among the enzymes in a proportion of 1:1:1.

**Table 9 foods-12-03754-t009:** Major phenolic compounds identified using HPLC-DAD in extracts obtained with MEAE-Tan-Cel-Pec before and after optimization.

Phenolic Compound	MEAE-Tan-Cel-Pec before Optimization * (mg/kg Pomace)	MEAE-Tan-Cel-Pec after Optimization ** (mg/kg Pomace)
Hydroxytyrosol	11.10 ± 4.65 ^b^	59.29 ± 2.97 ^a^
Protocatechuic acid	8.40 ± 0.49 ^a^	6.36 ± 1.24 ^a^
Chlorogenic acid	184.50 ± 99.65 ^a^	217.55 ± 15.19 ^a^
Tyrosol	8.70 ± 4.06 ^a^	3.51 ± 2.78 ^a^
Hydroxyphenylacetic	-	5.99 ± 2.88 ^a^
acid		
Vanillic acid	38.40 ± 15.64 ^a^	22.60 ± 6.01 ^a^
Caffeic acid	4.80 ± 0.47 ^a^	12.47 ± 2.19 ^a^
Vanillin	1.65 ± 2.27 ^a^	1.47 ± 0.43 ^a^
*p-*coumaric acid	11.70 ± 3.91 ^a^	20.55 ± 0.48 ^a^
Ferulic acid	24.04 ± 8.90 ^b^	140.88 ± 6.33 ^a^
*m-*coumaric acid	1.19 ± 0.23 ^a^	3.38 ± 0.91 ^a^
Elenolic acid	467.25 ± 120.24 ^b^	735.29 ± 10.80 ^a^

Values are presented as mean ± standard deviation. ^a,b^ Means within a row followed by different letters differ significantly (*p* ≤ 0.05) using Student’s unpaired *t*-test. * Microwave-assisted enzymatic extraction: 1% (*w*/*w*) enzyme mixture, 60 °C, 30 min of reaction under microwave irradiation, and 1:15 pomace/water ratio. ** Microwave-assisted enzymatic extraction under optimized conditions: 2.00% (*w*/*w*) enzyme mixture, 46 °C, 15 min of reaction under microwave irradiation, and 1:20 pomace/water ratio. -: not detected.

## Data Availability

The data used to support the findings of this study can be made available by the corresponding author upon request.

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
