# Peer review of "Optimizing the Integration of Microwave Processing and Enzymatic Extraction to Produce Polyphenol-Rich Extracts from Olive Pomace"

_foods, 2023, doi:10.3390/foods12203754_

Round 1

Reviewer 1 Report (Previous Reviewer 6)

According to changes done by Authors I can state that most of my suggestion has been taken into consideration. Although I understand the Authors point of view I still cannot agree with some statements as well as methodological approach. But the scientific paper should be the stage for discussion in order to develop the overall knowledge. According to that I designate the paper as ready for publishing. 

Author Response

  1. According to changes done by Authors I can state that most of my suggestion has been taken into consideration. Although I understand the Authors point of view I still cannot agree with some statements as well as methodological approach. But the scientific paper should be the stage for discussion in order to develop the overall knowledge. According to that I designate the paper as ready for publishing.
  • Response: The authors thank you for your review and acceptance of the publication of the manuscript.  

Reviewer 2 Report (Previous Reviewer 5)

The manuscript previously revised was carefully improved following the reviewer suggestions and comments. Some minor changes should be made:

1. Considering the question 8 and respective authors' response, please highlight the importance of exploring the effectiveness-cost benefits of microwave-enzymes-assisted extraction in future studies (as stated in authors' response).

2. Please revise the references following the journal guidelines. For example, in reference 24, reformat the scientific name of plant "Olea Europaea" in italic letter. Also, the journals names should be formatted in abbreviated form for all references (in reference 24, the journal name is in the full form, while in reference 25, it is in the abbreviated form).

3. Please consider improving the overall organization of the manuscript.

Author Response

The manuscript previously revised was carefully improved following the reviewer suggestions and comments. Some minor changes should be made:

  1. Considering the question 8 and respective authors' response, please highlight the importance of exploring the effectiveness-cost benefits of microwave-enzymes-assisted extraction in future studies (as stated in authors' response).
  • Response: The effectiveness-cost benefits from MEAE (microwave-enzymes-assisted extraction) mentioned by author’s was regarding the reaction time, energy costs and phenolic profile from obtained extracts. Once enzyme reactions improve the phenolic profile according to specificity and synergy of employed enzymes, in the other hand these reactions could take from 1 to 5 hours, under heat and agitation bath reactors. We demonstrated in this manuscript that it was possible to reach the same performance, using enzymes under microwave the reaction after only 15 minutes. This is probably a good reason to explore Microwave extraction in future studies.

  1. Please revise the references following the journal guidelines. For example, in reference 24, reformat the scientific name of plant "Olea Europaea" in italic letter. Also, the journals names should be formatted in abbreviated form for all references (in reference 24, the journal name is in the full form, while in reference 25, it is in the abbreviated form).
  • Response: The references were revised and formatted according to the suggestion.

  1. Please consider improving the overall organization of the manuscript.
  • Response: The overall organization of the manuscript was improved.

Reviewer 3 Report (Previous Reviewer 4)

The authors improved the manuscript well. Appropriate edits have been made. 

Author Response

The authors improved the manuscript well. Appropriate edits have been made.

  • Response: The authors would like to thank for the comments in order to improve the manuscript.

Reviewer 4 Report (Previous Reviewer 2)

I reviewed the manuscript title “Optimizing the integration of microwave processing and enzymatic extraction to produce polyphenol-rich extracts from olive pomace

There are many studies available in the literature using green extraction technologies for the recovery of phenolic compounds. This is not novel research. Authors performed optimization studies by changing the few parameters.

https://www.mdpi.com/2076-3921/12/6/1175

https://doi.org/10.1016/j.lwt.2020.110621 (authors tried to reply for this manuscript in previous version). However, this is highly correlated with LWT publication

https://doi.org/10.3390%2Ffoods11142002

https://doi.org/10.1021/acssuschemeng.0c09426

Minor language editing is required 

Author Response

I reviewed the manuscript title “Optimizing the integration of microwave processing and enzymatic extraction to produce polyphenol-rich extracts from olive pomace

There are many studies available in the literature using green extraction technologies for the recovery of phenolic compounds. This is not novel research. Authors performed optimization studies by changing the few parameters.

https://www.mdpi.com/2076-3921/12/6/1175

https://doi.org/10.1016/j.lwt.2020.110621 (authors tried to reply for this manuscript in previous version). However, this is highly correlated with LWT publication

https://doi.org/10.3390%2Ffoods11142002

https://doi.org/10.1021/acssuschemeng.0c09426

  • Response: We would like to reinforce that the innovation in this study was not the optimization process, but a combination of parameters. First of all, the use of olive pomace from North California producer is completely new and it is part of the effort to develop a local, new and unique plantation based on circular economy and have quality recognition in the market. The second parameter was the use of enzymes, as tannase (not commercial) under microwave conditions, which as far as we have noticed, it was not used neither pure or in combination with cellulase and pectinase under microwave conditions. The possibility to obtain an extract from pomace with high antioxidant potential and free of organic solvents is very promising to the food industry worldwide. These points were discussed in the first reviewer letter but probably was still not clear.

Reviewer 5 Report (Previous Reviewer 1)

The authors used Microwave- assisted enzymatic extraction of polyphenolic compounds from olive pomace. Three enzymes viz. tannase and a combination of Tan-Cel-Pec was used and processing protocols were optimized. The authors reported a synergistic effect of microwave and enzyme application in extracting phenolic compounds from olive pomace. An in-depth analysis of all polyphenolic compounds extracted from olive pomace also reported. Considering the huge amount of olive pomace produced globally every year, the extraction of these compounds could play a significant role in the circular economy and may have implications in the food industry.

The language is easy and clear to understand. The hypothesis is clear and sound.

I have my observations as follows-

Abstract: Perfect, at L26: Please what are the proportion of each enzyme in the Tan-Cel-Pec; they are equal or 2% each etc.

Keywords: May be arranged as Green extraction, enzymes, olive pomace, phenolic compounds. The phenolics are bioactive compounds and antioxidants, so they may be deleted.

Introduction: The introduction section provides sufficient background information and well justifies the need to take the study.

L46: please add if more recent data is available.

L55-69: The authors have justified the research work and are good for improving the hypothesis.

Results and discussion: Well-described and supported by suitable references.

Author Response

The authors used Microwave- assisted enzymatic extraction of polyphenolic compounds from olive pomace. Three enzymes viz. tannase and a combination of Tan-Cel-Pec was used and processing protocols were optimized. The authors reported a synergistic effect of microwave and enzyme application in extracting phenolic compounds from olive pomace. An in-depth analysis of all polyphenolic compounds extracted from olive pomace also reported. Considering the huge amount of olive pomace produced globally every year, the extraction of these compounds could play a significant role in the circular economy and may have implications in the food industry.

The language is easy and clear to understand. The hypothesis is clear and sound.

I have my observations as follows:

  1. Abstract: Perfect, at L26: Please what are the proportion of each enzyme in the Tan-Cel-Pec; they are equal or 2% each etc.
  • Response: The enzymes were applied in the proportion of 1:1:1, and 2% was total weight used. This information was included in the abstract (L24 – L25).

  1. Keywords: May be arranged as Green extraction, enzymes, olive pomace, phenolic compounds. The phenolics are bioactive compounds and antioxidants, so they may be deleted.
  • Response: The keywords were modified as suggested.

  1. Introduction: The introduction section provides sufficient background information and well justifies the need to take the study.
  • Response: Thank you for your review.

  1. L46: please add if more recent data is available.
  • Response: This information was updated according to FAOSTAT data.

  1. L55-69: The authors have justified the research work and are good for improving the hypothesis.
  • Response: Thank you for your opinion.

  1. Results and discussion: Well-described and supported by suitable references.
  • Response: Thank you for your opinion.

Round 2

Reviewer 4 Report (Previous Reviewer 2)

No suggestions were addressed. Authors must appropriately address the suggestions. Moreover, this manuscript lacks novelty. There are many publications are available in the literature 

Author Response

Reviewer #4:

“No suggestions were addressed. Authors must appropriately address the suggestions. Moreover, this manuscript lacks novelty. There are many publications are available in the literature”.

  • Response: Dear reviewer, thank you for your comments. We appreciate all the articles you sent us in the last review. Their similarity with our study are the use of emerging technologies (microwave and pressurized liquids) and the reuse of olive waste. However, none of them integrate microwave and enzymatic extraction.
  • A search in the Scopus database using the keywords "microwave", "enzyme" and “extraction” covers few articles and only one with olive pomace, previously published by our research group (https://doi.org/10.1016/j.lwt.2020.110621). The difference between this study and the one published in LWT was described in the manuscript (L79-L92).
  • We would like to confirm that the novelty of this study relies on the integration and optimization of the use of green technologies such as microwave and enzyme-assisted extraction (MEAE) to improve the extraction efficiency of bioactive phenolics from olive pomace while reducing processing time and costs. While these techniques have been evaluated isolated, the benefits of using both processing strategies simultaneously remain largely unexplored, with limited knowledge about the impact of key extraction fundamentals on processing efficiency. The primary of this study was to elucidate the role of essential extraction parameters in MEAE of phenolics from olive pomace. Additionally, this study aimed to pinpoint the optimal conditions that would effectively enhance the utilization of this promising extraction approach.

This manuscript is a resubmission of an earlier submission. The following is a list of the peer review reports and author responses from that submission.

Round 1

Reviewer 1 Report

The research manuscript by Macedo et al deals with the application of Microwave and Enzyme assisted extraction technologies for extracting bioactive compounds from oil pomace. The hypothesis is well explained and the language is easy to understand and read. The manuscript has significant recommendations. However, to further improve it, I have the following suggestions as-

       i.          Abstract: I would rather delete L13. The authors have mentioned the optimization and how these optimizing parameters were used for the study. Plz add 1-2 lines of experimental design (very brief)

     ii.          combined with pectinase and cellulase (MEAE-Tan-Cel-Pec), may be better written as cellulase and pectinase for a true indication of acronyms, plz correct.

   iii.          L26: may use readily available/ vast resources/agro-industrial byproducts/ sustainable resources in place of inexpensive materials for better support to the hypothesis

   iv.          Keywords: Have scope for improvement. I would suggest mentioning green extraction, olive pomace, enzymes, and bioactive compounds

     v.          L53-54: please also mention the low temp and processing conditions also help to get these bioactive ingredients without denaturation/ degradation of active principle etc

   vi.          A brief discussion on the choice of these enzymes would further improve the hypothesis.

  vii.          L97-98: parameters range was determined by preliminary trials and available literature. A shifting of para from L 187-197 here would help in better understanding the experimental design.

viii.          Result and Discussion: well-described results with suitable references except for some paras I found very long and can be easily concise such as the case of L247-263.

   ix.          Conclusion: Plz, should focus on the salient findings/ recommendations only and authors may avoid discussing experimental design details or background information here.

Thanks for giving me the opportunity to read your work. 

Author Response

Reviewer #1:

We are pleased to receive your comments and suggestions regarding our manuscript entitled “Optimization of combined microwave- and enzyme-assisted green extraction aiming to produce polyphenols-rich extracts from olive pomace”. We appreciate all the suggestions and observations, which helped us improve our manuscript. We have carried out corrections respecting all the suggestions. Details of these corrections and our responses to your comments are listed below and marked in the manuscript. We kindly hope you will find this revised version fully suitable for publication in the Journal Foods.

i. Abstract: I would rather delete L13. The authors have mentioned the optimization and how these optimizing parameters were used for the study. Plz add 1-2 lines of experimental design (very brief)

Response: The changes were performed as suggested.

ii. Combined with pectinase and cellulase (MEAE-Tan-Cel-Pec), may be better written as cellulase and pectinase for a true indication of acronyms, plz correct.

Response: Thanks for the observation. L13 has been changed as suggested.

iii. L26: may use readily available/ vast resources/agro-industrial byproducts/ sustainable resources in place of inexpensive materials for better support to the hypothesis.

Response: The change was performed as suggested.

iv. Keywords: Have scope for improvement. I would suggest mentioning green extraction, olive pomace, enzymes, and bioactive compounds

Response: Keywords were revised and changed as suggested.

v. L53-54: please also mention the low temp and processing conditions also help to get these bioactive ingredients without denaturation/degradation of active principle etc.

Response: The information was included on L58 and L63, as suggested.

vi. A brief discussion on the choice of these enzymes would further improve the hypothesis.

Response: We agree with the suggestion. The information was included from L65 to L71.

vii. L97-98: parameters range was determined by preliminary trials and available literature. A shifting of para from L 187-197 here would help in better understanding the experimental design.

Response: The information was included on L122 and L123, as suggested.

viii. Result and Discussion: well-described results with suitable references except for some paras I found very long and can be easily concise such as the case of L247-263.

Response: We agree that long paragraphs difficult the understanding. Thus, we summarized some discussions, as the case of L247-263 (now presented at L271-288).

ix. Conclusion: Plz, should focus on the salient findings/ recommendations only and authors may avoid discussing experimental design details or background information here.

Response: We agree with the suggestion. Conclusion was summarized and focused on the main findings.

Reviewer 2 Report

I reviewed the manuscript entitled, Optimization of combined microwave- and enzyme-assisted green extraction aiming to produce polyphenols-rich extracts from olive pomace.

There is a similar article published in LWT https://doi.org/10.1016/j.lwt.2020.110621 (Integrated microwave- and enzyme-assisted extraction of phenolic compounds from olive pomace). Moreover, many studies are available in the literature dealing with integrated extraction techniques for the extraction of phenolics from olive pomace. Likewise, optimization of techniques is a very old concept of research interest. Very few experiments and not much contribution to the field. In my opinion, the quality of the manuscript is not in line with the journal standards. 

Author Response

Dear reviewer, thanks for your comments. We understand your concern about the previous article published in LWT, however we confirm that this submission brings novelty and different goals from the article published in LWT.

The first article screened univariately factors to recovery phenolic compounds from olive pomace, applying hydroalcoholic or microwave extraction, combined or not with enzymes. The article evidenced the viability of integrating technologies and employing enzymes under microwave action, which had not been done yet for tannase. However, the conditions studied in the LWT article achieved low yields of phenolic compounds.

In the manuscript submitted to Foods, we applied optimization statistical design to enable process modeling for future scale-up. In this manuscript we evaluated the synergic effect of different factors and their optimal conditions to increase phenolics recovery. The optimized condition of MEAE increased total phenolic content in 95%, from 341.9 mg GAE/kg (obtained in LWT article) to 7110.6 mg GAE/kg (obtained in Foods manuscript by optimized MEAE-Tan). However, this current study was only possible after carrying out the previous article that selected variables and the range of each one.

Reviewer 3 Report

It seems to me that it is a good investigation, that it really contributes. I have some comments in lines 66 and 69 significant parameter should be replaced by factor. In figures 1 and 3, the contour plot should be eliminated, leaving only the response surface ones, they contribute more and they would remain with higher resolution. Tables 1,2,3 and 4 that are in the material and methods section contain results and are confusing, you should give them another design so that these results appear in the results section. The same happens with section 2.2 where the characterization is handled and results are observed.

Author Response

Dear reviewer, thanks for your revision. We appreciate all the suggestions and observations, which helped us improve our manuscript. We have carried out corrections respecting all the suggestions. Details of these corrections and our responses to your comments are listed below and marked in the manuscript. We kindly hope you will find this revised version fully suitable for publication in the Journal Foods.

  1. “Parameters” was replaced by factors at L83 and L86;
  2. We eliminated the contour plot in Figures 1 and 3, leaving only the response surface;
  3. Table 1, 2, 3 and 4 containing results were placed in results section. Tables of material and methods section got another design.
  4. Results of 2.2 section were placed in results section.

Reviewer 4 Report

The task of the study was to find conditions that would allow obtaining polyphenol enriched extracts from olives. For this, various approaches were used, including both rather traditional ones (temperature, duration of exposure, microwave), and various enzymes (separately and in combination), which ensure the breakdown of plant material.

The manuscript presents a large amount of experimental material, which is discussed in detail. The conclusions on the work done are quite standard, although there is a certain novelty. Interesting differences in the composition of polyphenols, as well as the antioxidant activity of the extracts.

Remarks and comments are reflected in the text of the manuscript.

The authors should pay attention to the statistical processing of data, the assessment of the reliability of differences between the options, as well as the adequacy of the interpretation of the data obtained.

Reviewer 5 Report

The manuscript entitled "Optimizing the integration of microwave processing and enzymatic extraction to produce polyphenol-rich extracts from olive pomace" aims to optimize the extraction of phenolic compounds from olive pomace by an integrated approach of two green processes, namely microwave and enzyme-assisted extractions. The paper seems interesting and fulfills an opening field of research. However, some questions/comments should be addressed :

1. Why did the authors use these enzymes (tannase and a combination of tannase, celullase and pectinase) in the extraction of phenolics' olive pomace? Please justify it.

2. In introduction, please highlight the novelty of the study to the research field and relevance for practical applications (lines 83-94).

3. In methods' section, please include more details about the collection of olive pomace in subsection 2.2. (lines 105-107).

4. In methods' section, please describe briefly the extraction procedure by indicating the equipment and how were the extracts treated after extraction in the subsection 2.3. (lines 108-113).

5. Why did the authors selected these extraction conditions for the optimization studies (lines 114-148)? Please justify it.

6. In table 2, why were just these two factors studied for CCRD of MEAE-Tan (without temperature factor)? Please clarify it.

7. In methods section, please indicate the statistical tests applied in subsection 2.7. (lines 211-213).

8. In discussion, consider to discuss the effectiveness-cost benefits of applying microwave-enzymes-assisted extraction in comparison to other green extraction technique. 

9. Please consider including a list of abbreviations in the beginning of the manuscript.

10. Figures 2 and 4: Missing the error bars in the graphs. Please add them. Also, the statistics is missing and needs to be included.

11. Tables 7 and 9: The statistical analysis is missing in the table.

12. Discussion: Please compare the results with previous literature on olive pomace or similar matrices.

13: In conclusion, consider discussing further perspectives in the research field and potential industrial applications.

Minor comments:

- Line 132: Please consider using full names in the subtitles instead of abbreviations.

- Line 136: Add a space between point and "Based".

- Line 267: "synergistic"? Please revise it.

- Line 299: Delete the extra point after "processing time".

- Line 306: Correct "significant".

- Lines 321-324: Please add at least one reference to support this statement.

- Line 350: Correct "extractability".

- Line 354: Correct "yielding".

- Lines 357-359: Please revise the sentence.

- Figures 1, 2, 3, 4: Please improve the resolution of figures.

- Line 408: Add "the" before "highest".

- Lines 414-418: Please add references to support these statements.

- Line 462: Add a point after the reference "[38]".

- Format "p" and "m" from "p-coumaric acid" and "m-coumaric acid" in italic letter. Revise it along the manuscript.

- Lines 500-505: Please add references to support these statements.

Reviewer 6 Report

The paper looks interesting by mean of main goals including the utilization of olive pomace, one of the most important side product in countries producing olive oil. Unfortunately there is not enough data presented in the paper to fully understand the mechanism of proposed extraction. First of all there is no information on microwave power used for the experiment. This data is crucial for optimization itself as well for understanding phenomena. The second unknown factor is a geometry of vessel used for experiments and the way how microwaves are distributed thorough the sample. What type of microwave extractor was used? Monomode or multimode? In case of multimode system information on microwave power control should be shown. It is continuous or periodical. In case of periodic action any kinetic study as well as “temperature” of the process are not known. The next thing is the lack information on temperature measurement in microwave assisted process. In case of slurry/liquid system it is important to know if the temperature was measured by means of pyrometer (surface temperature – useless for planned experiments) or e.g. fiber optic employing semiconductors (very useful for planned experiments). Additionally “temperature” measured in experiments does not correspond to real temperature inside the sample. The local overheating i.e. hot spots probably caused differentiation in extraction mechanism.  To say the truth authors does not propose or cited any mechanism. There is also no comparison to conventional extraction in the same “temperature”/time/enzyme conditions. So why authors conclude on shorter extraction time and more environment friendly methods they developed when there is no comparison to classical extraction, no data on real power needed to conduct the process and in fact the even average temperature of the process is unknown. In the paper there is no information/graphs showing the temperature rate and no power profile can be found. Without all this information obtained data seems to be useless.

Round 2

Reviewer 2 Report

Optimization studies are already available in the literature. Changing one parameter or addition of enzymes will not contribute to the field. In my opinion, the manuscript still has novelty and weak scientific quality and thus I remain to recommend rejection.